# The Mechanism of Osteoprotegerin-Induced Osteoclast Pyroptosis In Vitro

**DOI:** 10.3390/ijms24021518

**Published:** 2023-01-12

**Authors:** Jiaqiao Zhu, Yonggang Ma, Jie Wang, Yangyang Wang, Waseem Ali, Hui Zou, Hongyan Zhao, Xishuai Tong, Ruilong Song, Zongping Liu

**Affiliations:** 1College of Veterinary Medicine, Yangzhou University, Yangzhou 225009, China; 2Joint International Research Laboratory of Agriculture and Agri-Product Safety, The Ministry of Education of China, Yangzhou University, Yangzhou 225009, China; 3Jiangsu Co-Innovation Center for Prevention and Control of Important Animal Infectious Diseases and Zoonoses, Yangzhou 225009, China

**Keywords:** osteoprotegrin, osteoclast, pyroptosis, GSDMD

## Abstract

Osteoprotegerin (OPG) is a new member of the tumor necrosis factor (TNF) receptor superfamily, which can inhibit the differentiation and activity of osteoclasts by binding to nuclear factor kappa B receptor activator (RANK) competitively with nuclear factor kappa B receptor activator ligand (RANKL). The previous experiments found that OPG can induce apoptosis of mature osteoclasts in vitro, which can inhibit the activity of mature osteoclasts, thereby exerting its role in protecting bone tissue. In addition, pyroptosis is a new type of cell death that is different from apoptosis. It is unclear whether OPG can induce mature osteoclast pyroptosis and thereby play its role in protecting bone tissue. In this study, the results showed that compared with the control group, the survival rate of osteoclasts in the OPG group was significantly reduced, and the contents of IL-1β, IL-18, and LDH in the supernatant both increased. Many osteoclast plasma membranes were observed to rupture in bright fields, and OPG induced loss of their morphology. Flow cytometry was used to analyze the pyroptosis rate; OPG significantly increased the osteoclast pyroptosis rate. To further reveal the mechanism of OPG-induced osteoclast pyroptosis, we examined the expression level of pyroptosis-related genes and proteins, and the results found that OPG increased the expression of NLRP3, ASC, caspase-1, and GSDMD-N compared with the control group. In summary, OPG can induce osteoclast pyroptosis, and its mechanism is related to the expression levels of ASC, NLRP3, caspase 1 and GSDMD, which were included in the classical pathway of pyroptosis.

## 1. Introduction

Osteoclasts are tissue-specific macrophages that are multinucleated, differentiated from monocytes/macrophages, and mainly exist at or near the bone surface [1]. Osteoclasts have the function of bone resorption, which can attach to the bone surface, form a closed zone, and digest the mineral and protein components of the bone through acidic substances and lysosomal enzymes secreted by the folds [2]. Therefore, it plays an important role in the pathogenesis of bone remodeling and bone metabolism-related diseases. Osteoprotegerin (OPG) is a new member of the TNF receptor superfamily, as a decoy receptor for RANKL, which can inhibit the differentiation and activation of osteoclast precursors by competitively binding RANKL with RANK [3]. It was found that OPG transgenic mice showed osteosclerosis, and OPG knockout mice showed severe osteoporosis [4]. In vitro results showed that OPG also inhibited the maturation and differentiation of osteoclast precursors and inhibited the activity of mature osteoclasts [5]. The study also found that the cessation of estrogen production in postmenopausal women leads to bone loss and osteoporosis in the elderly. However, OPG treatment reverses the increase in osteoclast numbers and bone loss [6]. In addition, we also found that OPG can induce apoptosis of mature osteoclasts in vitro through the Fas/Fasl receptor pathway [7]. Therefore, OPG may exert its protective effect on bone tissue in other ways than the competitive binding of RANKL.

Pyroptosis, discovered in the 1980s, is a new type of cell death that is different from apoptosis, characterized by cell membrane rupture and the release of inflammatory factors [8]. When pyroptosis occurs, the morphological changes of the cells are between apoptosis and necrosis, and a series of inflammatory reactions will be presented, including the formation of 1.1–2.4 nm pores on the cell membrane, intracellular K^+^ outflow, extracellular Na^+^ Ca^2+^ influx, and a massive release of cell contents and proinflammatory factors. Studies have found that pyroptosis is related to the pathogenesis of major diseases such as diabetes [9], tumors [10], and arthritis [11]. Therefore, pyroptosis has become a new target of many drugs and has attracted extensive attention from researchers. In the study of bone metabolism, monocytes, macrophages, and dendritic cells are commonly used as cell models to study pyroptosis. However, the molecular mechanism of pyroptosis of osteoclasts has not been reported. NLRP3 inflammasome activation is a key step in classical pyroptosis; NLRP3 inflammasome activation cleaved pro-caspase-1, pro-IL-1β/18, and GSDMD were digested by activated caspase-1, and the N-terminus of GSDMD protein was oligomerized and perforated in the cell membrane, which caused the release of intracellular inflammatory factor IL-1β/18 and the influx of extracellular fluid, eventually leading to cell swelling, rupture, and death. In the study of osteoclast apoptosis induced by OPG, our research group found that OPG can activate the caspase-3 pathway to cause cell apoptosis [7], but whether the caspase-1-mediated pyroptosis pathway is involved in osteoclast pyroptosis induced by OPG remains unknown. Based on the above understanding, this study investigated the molecular mechanism of osteoclast pyroptosis induced by OPG and provided new ideas for the treatment of clinical bone metabolic diseases.

## 2. Results

### 2.1. Effect of OPG on Survival Rate and Cell Membrane Integrity of Osteoclasts

After treating osteoclasts with OPG at different concentrations (0, 40, 80, 100 ng/mL) for 12 h, the survival rate of osteoclasts was detected by CCK-8. The survival rate of osteoclasts was significantly decreased as compared to the control group (Figure 1A). Next, the LDH cytotoxicity assay kit was used to detect changes in LDH activity in the culture supernatant. As shown in Figure 1B, compared with the control group, the LDH activity in the culture supernatant of the OPG (40 ng/mL) treatment group showed no significant change, and the LDH activity in the culture supernatant of the OPG (80 ng/mL) treatment group showed a significant increase, indicating that OPG promoted the release of LDH in osteoclasts. As shown in Figure 1C, scanning electron microscopy (SEM) was used to observe the surface morphology of osteoclasts. There were obvious holes on the surface of osteoclasts in the OPG-treated group, indicated by red arrows. We further observed the morphological changes of the osteoclast in each group under the light microscope. As shown in Figure 1D, osteoclasts in the treatment group underwent morphological changes of different degrees. The cell membrane of osteoclasts in the control group remained intact, and the cell morphology was normal. With the increase in OPG concentration, the number of dead osteoclasts gradually increased, and the dead cells experienced swelling, plasma membrane rupture, and cell morphology loss. In the OPG (100 ng/mL) treatment group, almost all osteoclasts died and their cell morphology disappeared. As shown in Figure 1E, TRAP staining further suggested that OPG destroys the osteoclast morphology. These results indicated that OPG destroys the morphology structure of osteoclast.

### 2.2. Effects of OPG on Death Rate and the Release of Inflammatory Factors of Osteoclasts

Flow cytometry was used to detect the pyroptosis rate of osteoclasts. After different concentrations of OPG treatment for 12 h, the osteoclast was stained with PI and FITC; in this experiment, if osteoclasts were double-stained, they were mainly distributed in the Q2 quadrant and were defined as having pyroptosis. Therefore, as shown in Figure 2A, the pyroptosis rate of osteoclasts significantly increased with increasing OPG concentration, compared to the control group, but OPG (40 ng/mL) had no significant effect on the pyroptosis rate of osteoclasts. These results indicated that OPG could induce pyroptosis of osteoclasts. As shown in Figure 2B, after treatment with OPG (0, 40, 80, 100 ng/mL) for 12 h, compared with the control group, the contents of IL-1β and IL-18 in the culture supernatant were increased. The results revealed that OPG promotes the release of inflammatory factors in osteoclasts.

### 2.3. Effect of OPG on Transcription Levels of Pyroptosis Pathway-Related Genes of Osteoclasts

After treating osteoclasts with OPG at different concentrations (0, 40, 80, and 100 ng/mL) for 12 h, qRT-PCR was used to detect the transcription levels of pyroptosis-related genes (ASC, caspase-l, GSDMD, NLRP3, IL-18, and IL-1β). As shown in Figure 3, the results showed that the m-RNA transcription level of ASC was not significantly different from that of the control group under 40 ng/mL OPG treatment but significantly increased after 80 ng/mL and 100 ng/mL OPG treatment. Compared with the control group, the m-RNA transcription levels of caspase-l, GSDMD, NLRP3, IL-18, and IL-1β were significantly increased under 40, 80, and 100 ng/mL OPG treatment. These results showed that OPG increased the expression levels of pyroptosis-related genes.

### 2.4. Effects of OPG on Expression Levels of Pyroptosis Pathway-Related Proteins of Osteoclasts

As shown in Figure 4A,B, after 12 h treatment of osteoclasts with OPG at different concentrations (0, 40, 80, 100 ng/mL), protein expression levels of caspase-l, cleaved caspase-l, NLRP3, and ASC were significantly increased with increasing OPG concentration compared with the control group. As shown in Figure 4C, compared with the control group, the protein expression of inflammatory factors IL-1β, and IL-18 were also obviously increased with the increase in OPG concentration. As shown in Figure 4D, the expression of GSDMD-N was increased compared with the control group. Their results indicate that OPG increases the expression of pyroptosis proteins on osteoclasts.

## 3. Discussion

Osteodystrophy is caused by an imbalance between bone resorption and bone formation; bone resorption is greater than bone formation. Osteoclasts are specific multinucleated macrophages with bone resorption function [12]. Osteoblasts, another important cell in bone tissue, are closely related to each other and work together to maintain bone homeostasis. Under normal physiological conditions, osteoclasts constantly destroy or absorb bone tissue and then are replaced by osteoblasts, a continuous bone remodeling process. When the balance between the two is disrupted, the body can develop diseases such as osteoporosis. Therefore, osteoclasts have always been an important target in treating osteodystrophy diseases [13].

Since the discovery of OPG, almost all studies on OPG have focused on its effects on osteoclast formation and bone resorption. In recent years, some researchers have explained the effect of OPG on osteoclast survival and apoptosis [5,14,15]. In addition, OPG may induce other death modes of mature osteoclasts to play its protective role. In this study, the survival rate of osteoclasts in the OPG treatment group was significantly decreased, and the cytoplasmic LDH was released into the supernatant in large quantities. The research also found that all osteoclasts in the experimental group showed plasma membrane rupture, cell morphology changes, or even a loss of cell morphology. Among them, almost all osteoclasts in the 100 ng/mL OPG treatment group died, cell membranes ruptured, cell boundaries disappeared, and only the outlines of cells remained. The research showed that pyroptosis is characterized by cell swelling and cell membrane rupture [16]. Therefore, pyroptosis is likely to play an essential role in the bone protection of OPG.

Moreover, like apoptotic cells, Annexin-V can stain pyroptosis cells. However, the difference is that pyroptosis cells can be stained with propyl iodide (PI) because the plasma membrane integrity of pyroptosis cells is damaged, while apoptotic cells keep the membrane integrity and will not be stained with propyl iodide (PI). Therefore, we used AnnexinV-FITC and PI double staining to screen pyroptosis cells. The results found that the AnnexinV-FITC and PI-positive cells were increased in the OPG treatment group. The previous study [7] found that OPG (0, 20, 40, 80 ng/mL) induced precursor osteoclast apoptosis; in this study, we found that 100 ng/mL OPG treated mature osteoclasts for 12 h, which induced osteoclast pyroptosis. This difference is mainly caused by the difference of OPG treatment time and concentration. In addition, we think that after OPG treatment, apoptosis and pyroptosis may be simultaneous low-concentration OPG-induced osteoclast apoptosis and high-concentration OPG-induced pyroptosis. In all, in different conditions, OPG may play a different role in osteoclast. In this study, we also showed that pyroptosis occurs with the release of inflammatory cytokines IL-1β and IL-18 [17]. In this study, we found that the content of IL-1β and IL-18 in the supernatant of the medium increased significantly in the experimental group and showed a dose-dependent effect. Many studies showed that in the absence of stimulation, the full-length GSDMD remains intact, with the N-terminal (GSDMD-N) and C-terminal (GSDMD-C) regions interacting, and this self-suppressed conformation is divided into GSDMD-N and GSDMD-C, followed by GSDMD-N after effective cutting by caspase-1 or caspase-11 oligomerization occurs and translocations to the cell membrane in its oligomeric form, causing changes in cell membrane permeability and further inducing cell rupture [18,19].

Interestingly, we also found the expression level of GSDMD-N in the treatment group was significantly increased, indicating that OPG treatment could upregulate the expression level of GSDMD-N protein in osteoclasts, finally inducing pyroptosis of osteoclasts. In addition, it was also found that caspase-1 also cleaved the precursors of IL-lβ and IL-18, making them mature IL-lβ and IL-18, usually via the NLRP3/ caspase-1 pathway, which mediates the formation and secretion of mature IL-lβ/IL-18 [20,21,22]. Studies have shown that NLRP3 inflammasome can be activated by different stimuli to collect ASC and form a complex with caspase-1 through which pro-caspase-1 is activated, which is defined as the classical pathway of pyroptosis [23]. Our results showed that the transcription levels of NLRP3, ASC, and caspase-l genes and their protein expression levels in OPG-treated osteoclasts increased. Therefore, these results indicated that OPG could upregulate the expression of NLRP3, ASC, and caspase-l genes after treatment of osteoclasts and thus activate the pyroptosis classical pathway.

## 4. Materials and Methods

### 4.1. Experimental Animals

Balb/c mice were supplied by the Comparative Medicine Center of Yangzhou University (Yangzhou, China).

### 4.2. Reagents

α-MEM and Fetal bovine serum (FBS) were purchased from Gibco (Gibco, Carlsbad, CA, USA); OPG, M-CSF, RANKL, and OPG were obtained from R&D systems (R&D, Minnesota, USA). RIPA pyrolysis liquid was supplied from Biyuntian (Biyuntian Biotechnology Research Institute Shanghai, China). All antibodies were purchased from CST (CST, Boston, MA, USA). 

### 4.3. Cell Culture

Bone marrow cells were obtained from the femurs and tibias of male Balb/c mice at 4 weeks old, and bone marrow macrophages (BMMs) were seeded in 6-well plates for 8 × 10^6^/mL, and were cultured in α-MEM) containing 10% FBS in the presence of 30 ng/mL M-CSF and 60 ng/mL RANKL for 4 days. After 4 days, mature osteoclasts were treated with OPG (0, 40, 80, 100 ng/mL) for 12 h.

### 4.4. LDH, IL-18, and IL-1β Assay

BMMS were seeded in the 12-well plates in the presence of M-CSF and RANKL for 8 × 10^6^/mL. After 4 days, different concentrations of OPG were treated for 12 h, and the medium supernatant was collected and centrifuged at 1000 rpm. The medium supernatant concentration was quantified with BCA kit. The samples were measured using IL-18, IL-1β, and LDH ELISA kits (Mlbio, Shanghai, China). Experiments were conducted according to the kit protocol. The OD value of each well was measured at 450 nm wavelength.

### 4.5. Cellular Viability

BMMs were cultured in the 96-well plates in the presence of M-CSF and RANKL, and after different concentrations of OPG treatment, cellular viability was assayed by cell counting kit-8 (CCK-8) assay (Vazyme, Nanjing, China).

### 4.6. Scanning Electron Microscope (SEM)

BMMs were cultured in the presence of M-CSF and RANKL for 5 days. After OPG treatment, the osteoclasts were fixed in 2.5% glutaraldehyde solution. To observe their morphology, the specimens were coated with gold using an SCD 500 sputter-coater (Leica, Wetzlar, Germany) and examined using a Hitachi S-4800 field-emission environmental scanning electron microscope (Hitachi, Tokyo, Japan).

### 4.7. Flow Cytometry Enumeration

BMMs were seeded in 6-well plates in the presence of RANKL and M-CSF after OPG treatment. Cells were collected and stained with the Annexin V-FITC/PI kit (Vazyme, Nanjing, China). The mature osteoclast death rate was analyzed by flow cytometry enumeration (BD, Franklin Lake, NJ, USA).

### 4.8. TRAP Staining

Mature osteoclasts were stained for TRAP kit (St. Louis, MO, USA). Multinucleated TRAP-positive cells containing at least three nuclei were identified as osteoclast-like cells using an inverted phase-contrast microscope (Leica, Wetzlar, Germany).

### 4.9. Western Blot

After mature osteoclasts were treated with OPG for 12 h, the cells were washed with cold PBS and lysed in RIPA buffer on ice. After 30 min, the cells were collected and centrifugated at 12,000 rpm for 10 min at 4 °C. Then, the protein concentration was quantified by BCA kit (Yeasen, Shanghai, China). Equivalent amounts of protein were separated on 12% SDS-PAGE and transferred into NC membranes. Each membrane was blocked for 2 h using 5% skim milk and then incubated with primary antibody and then with secondary antibodies. Immunoreactive proteins were detected using chemiluminescence (NCM Biotech, Su Zhou, China). The detected bands were quantified using ImageJ Software (NIH, Bethesda, MD, USA).

### 4.10. Statistical Analysis

The data were analyzed statistically and expressed as mean ± standard deviation (SD) from at least three independent experiments. GraphPad Prism 6 software (GraphPad Software lnc., La Jolla, CA USA) analyzed the data using a one-way analysis of variance ((ANOVA) (Scheffe’s SF test). P values less than 0.05 indicated a significant difference.

## 5. Conclusions

In this study, osteoclasts were treated with OPG for 12 h; OPG could affect the survival of the osteoclasts. In addition, OPG treatment induced pyroptosis of the osteoclasts by upregulating the transcription levels and protein expression levels of genes related to pyroptosis classical pathways such as ASC, NLRP3, caspase-1, and GSDMD. In all, this study preliminarily revealed the mechanism of OPG-induced pyroptosis of osteoclasts. 

## Figures and Tables

**Figure 1 ijms-24-01518-f001:**
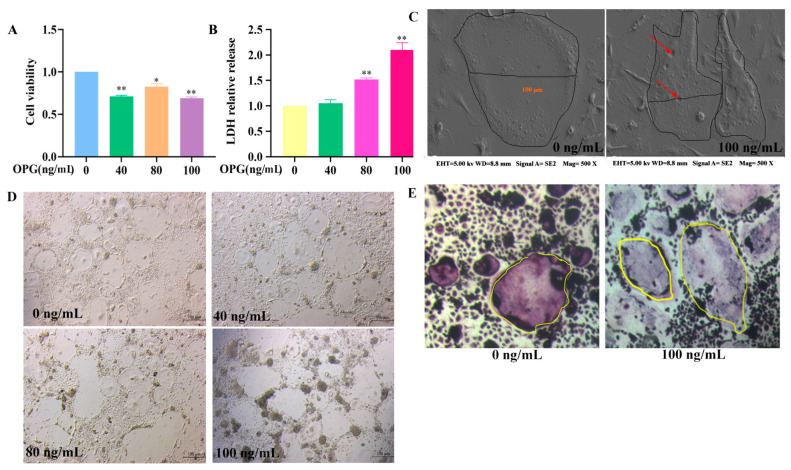
Effect of OPG on survival rate and cell membrane integrity of osteoclasts. (**A**) Cell viability was analyzed by CCK-8. (**B**) LDH release was examined by ELISA. (**C**) SEM was used to observe the morphology of osteoclasts; magnification is 500 ×. (**D**) osteoclasts’ morphology was observed by microscope; scale bar =100 μm. (**E**)—Mature osteoclasts were stained with TRAP; scale bar = 50 μm. (mean ± SD, n = 3, * *p* < 0.05, ** *p* < 0.01).

**Figure 2 ijms-24-01518-f002:**
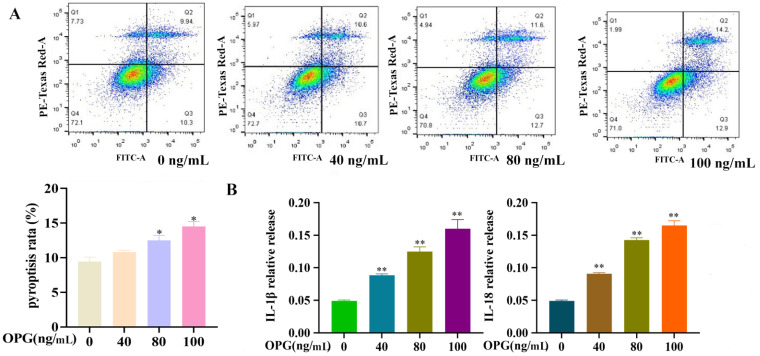
Effects of OPG on death rate and the release of inflammatory factors of osteoclasts. (**A**) Osteoclasts’ death rate was analyzed by FCM. (**B**) IL-18 and IL-1β were examined by ELISA. (mean ± SD, n = 3, * *p* < 0.05, ** *p* < 0.01).

**Figure 3 ijms-24-01518-f003:**
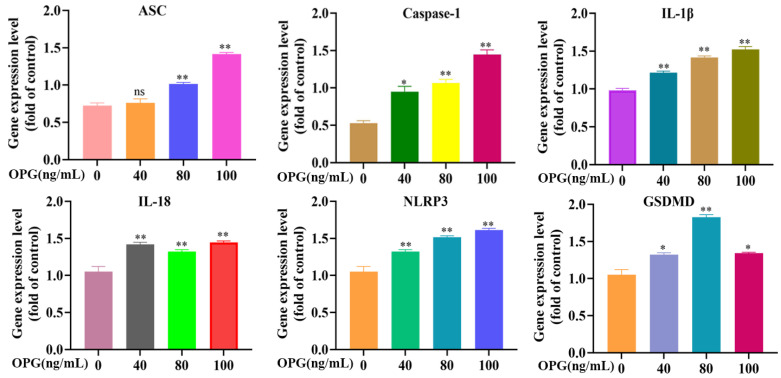
Effect of OPG on transcription levels of pyroptosis pathway-related genes of osteoclasts. ASC, Caspase-1, IL-18, IL-1β, NLRP3, and GSDMD were quantified by q-PCR. (mean ± SD, n = 3, * *p* < 0.05, ** *p* < 0.01).

**Figure 4 ijms-24-01518-f004:**
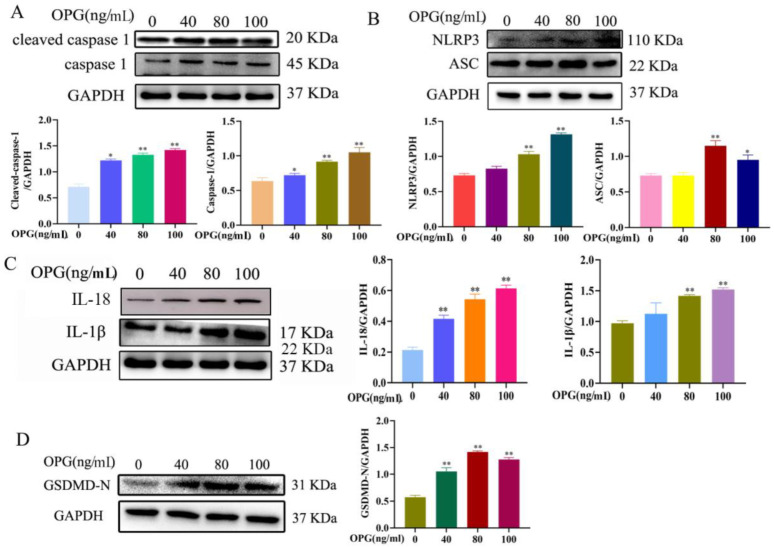
Effects of OPG on expression levels of pyroptosis pathway-related proteins of osteoclasts. (**A**). The expression of cleaved caspase 1 and caspase 1; (**B**). The expression of NLRP3; (**C**). The expression of IL-18 and IL-1β; (**D**). The expression of GSDMD-N. ASC, Caspase-1, IL-18, IL-1β, NLRP3, and GSDMD were quantified by Western blot. (mean ± SD, n = 3, * *p* < 0.05, ** *p* < 0.01).

## Data Availability

Not applicable.

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
