# Peer review of "The Mechanism of Osteoprotegerin-Induced Osteoclast Pyroptosis In Vitro"

_ijms, 2023, doi:10.3390/ijms24021518_

Round 1

Reviewer 1 Report

In the proposed manuscript, authors investigated the effect of different conc. of OPG on osteoclast-like cell pyroptosis. Therefore, cells were isolated from mice and differentiated to osteoclast-like cells. Different conc. of OPG were used to treat cells after differentiation. Key players of inflammation and pyroptosis were investigated in further experiments.

Herein, I would like to address the following major comments:

Title:

The title "The mechanism of osteoprotegerin-induced mice osteoclast pyroptosis" needs to be changed because of several flaws: the english style is not good and it does not reflect the study, since the authors did not inject OPG to induce osteoclast pyroptosis. The authors just isolated cells from mice and treated the cells with OPG. 

Materials and Methods:

General comment: The description in the M&M part is just spare. It is difficult to follow, which treatment was used for which further experiment. I kindly ask the authors to rewrite this part in a sensefull manner. Moreover, why did the authors use mice or mouse primary cells?

Line 81-83: I kindly ask the authors to rewrite this part, because it is difficult to understand that the animals were just sacrificed for cell isolation, which is a huge difference concerning ethics. 

Line 86-88: In this part of the materials and methods, something is missing, especially see line 87 and 88.

Line 94: Use a capital for "Bone marrow...."

Line 121: For western blot analysis, how long did the authors treat the osteoclasts with OPG?

Line 126: The "E" is missing in "12 % SDS-PAGE"

Results:

Line 139: Here, the methods part is more clear than in the actual M&Ms part, please include more details in the M&M section.

Line 144 and 148: Please use a space here "Fig1B" and "Fig1C"

Line 163, Figure 1: I think the authors mean "ELISA"

Figure 1D: Please enlarge the letters describing the concentrations used

Line 168-170: This sentence is not understandable, please rewrite this part "Flow cytometry was used to detect the pyroptosis rate of osteoclasts, different concentrations (0, 40, 80, 100ng/ml) 12h after OPG treatment, and the cells distributed in Q2 quadrant were pyroptosis cells, as shown in Fig 2A"

Discussion:

Line 244-246: To confirm the following statement "The results found that 244 the AnnexinV-FITC and PI-positive cells were increased in the OPG treat- 245 ment group.", hard numbers (quantification) is needed. 

Line 275-276: The following statement "In this study, the osteoclasts were treated with OPG for 12 h; OPG could affect the survival of osteoclasts in a dose-dependent manner" is not entirely true, because cell viability results were not dose-dependent. 

Author Response

Question 1:

Title:The title "The mechanism of osteoprotegerin-induced mice osteoclast pyroptosis" needs to be changed because of several flaws: the english style is not good and it does not reflect the study, since the authors did not inject OPG to induce osteoclast pyroptosis. The authors just isolated cells from mice and treated the cells with OPG. 

Response: we have modified.

Question 2:

Materials and Methods:

General comment: The description in the M&M part is just spare. It is difficult to follow, which treatment was used for which further experiment. I kindly ask the authors to rewrite this part in a sensefull manner. Moreover, why did the authors use mice or mouse primary cells?

Response: Materials and Methods part have rewrote. In this study, we used BMMs to induce osteoclast, which is precursor osteoclast, and can really reflect osteoclast function. Therefore, we do not choose RAW264.7 to induce osteoclast.

Question 3:

Line 81-83: I kindly ask the authors to rewrite this part, because it is difficult to understand that the animals were just sacrificed for cell isolation, which is a huge difference concerning ethics. 

Line 86-88: In this part of the materials and methods, something is missing, especially see line 87 and 88.

Line 94: Use a capital for "Bone marrow...."

Line 121: For western blot analysis, how long did the authors treat the osteoclasts with OPG?

Line 126: The "E" is missing in "12 % SDS-PAGE"

Response: we have solved above question.

Question 4:

Results:

Line 139: Here, the methods part is more clear than in the actual M&Ms part, please include more details in the M&M section.

Line 144 and 148: Please use a space here "Fig1B" and "Fig1C"

Line 163, Figure 1: I think the authors mean "ELISA"

Figure 1D: Please enlarge the letters describing the concentrations used

Line 168-170: This sentence is not understandable, please rewrite this part "Flow cytometry was used to detect the pyroptosis rate of osteoclasts, different concentrations (0, 40, 80, 100ng/ml) 12h after OPG treatment, and the cells distributed in Q2 quadrant were pyroptosis cells, as shown in Fig 2A"

Response: we have solved above question.

Question 5:

Discussion:

Line 244-246: To confirm the following statement "The results found that 244 the AnnexinV-FITC and PI-positive cells were increased in the OPG treat- 245 ment group.", hard numbers (quantification) is needed. 

Response: we have quantified.

Question 6:

Line 275-276: The following statement "In this study, the osteoclasts were treated with OPG for 12 h; OPG could affect the survival of osteoclasts in a dose-dependent manner" is not entirely true, because cell viability results were not dose-dependent. 

 Response: we have modified.

Reviewer 2 Report

The present study analyse “The mechanism of osteoprotegerin-induced mice osteoclast py roptosis”

Comments:

Comment 1. Authors cultured bone marrow macrophages with 30 ng/ml M-CSF and 60 ng/ml RANKL.The all methodology used in the cell culture experiments needs to be detailed: number of plated cells, culture time before OPG treatment; the medium composition during the OPG treatment, …. For instance, the levels of OPG used were never mentioned in the Materials and Methods. The OPG concentration range (0, 40, 80, 100 ng/mL) was mentioned for the first time in the section “Results”.

Comment 2. Levels of LDH, IL-18 and IL-1β assay in the medium supernatant were not normalized to the cell content in each experimental condition. Plating the same number of cells will give different cell populations in the various experimental conditions – thus, some sort of normalization is needed (to total protein content, DNA content, …)

Comment 3. SEM images (Figure 1) are really of poor quality – they do not allow the detailed description of the morphology and alterations described the authors. The same is observed with the light microscopy images – they do not allow to identify multinucleated cells. Authors need to prove that they have osteoclastic cells – i.e., performing immunostaining of the cytoskeleton and nucleus allows the visualization of the typical actin rings and the multiple nucleus. Even histochemical TRAP staining (a very basic staining) would provide useful information. And, why TRAP activity was not determined – it is a very basic biochemical method of a osteoclastic marker.

Comment 4. In a previous work reported by the research team (ref. 7), authors used a similar experimental setup for the obtention of the osteoclasts (bone marrow from male Balb/cJ mice) and also analysed the effect of OPG (in a similar range - 0, 20, 40, 80 ng/mL, apparently, 24 h treatment) in the dead rate. In this past study, they used the term “apoptosis” (Q2 quadrant) but in the present work, with the same protocol, the term used was “pyroptosis” (Q2 quadrant). They analysed distinct underlying mechanisms for the cell dead rate – “Osteoprotegerin Induces Apoptosis of Osteoclasts and Osteoclast Precursor Cells via the Fas/Fas Ligand Pathway” (Ref 7) and “The mechanism of osteoprotegerin-induced mice osteoclast pyroptosis” (present study).

Cells in Q2 quadrant may behave similarly in flow cytometry using Annexin V-FITC/PI kit (cells in late apoptosis / pyroptosis), and they can be distinguished by complementary information and underlying mechanisms. But relating the two studies using a similar protocol, OPG may be inducing both cell dead processes? Surprisingly, in the “Discussion” section, authors never mentioned this previous study, and discuss/relate the results of both works.

Why these two studies were not discussed in an integrative way? 

Comment 5. The Conclusions of the two studies reflect the obtained results:

-        Previous study (ref. 7): In conclusion, our investigations reveal that OPG promotes apoptosis of OCs and OPCs by activating the Fas/FasL pathway. In addition, OPG-induced Fas/FasL activation regulates osteoclast apoptosis using any cell-type-specific signaling pathways. Such pathways could provide novel designs of osteoclast-specific therapeutic interventions in bone disorders. Therefore, future research is required on the precise Fas-mediated intracellular signaling pathways in osteoclast apoptosis induced by OPG.

-       Present study: In this study, the osteoclasts were treated with OPG for 12 h; OPG could affect the survival of osteoclasts in a dose-dependent manner. In addition, OPG treatment induced pyroptosis of osteoclasts by up-regulating the transcription levels and protein expression levels of genes related to pyroptosis classical pathways such as ASC, NLRP3, caspase-1, and GSDMD. In all, this study preliminarily revealed the mechanism of OPG-induced pyroptosis of osteoclasts, providing a theoretical basis for the treatment of bone metabolic diseases.

 In a translational point of view - novel insights of osteoclast-specific therapeutic interventions in bone disorders – stressed by the two studies, what will be the difference in the clinical outcome?

Author Response

Question 1:

Comment 1. Authors cultured bone marrow macrophages with 30 ng/ml M-CSF and 60 ng/ml RANKL.The all methodology used in the cell culture experiments needs to be detailed: number of plated cells, culture time before OPG treatment; the medium composition during the OPG treatment, …. For instance, the levels of OPG used were never mentioned in the Materials and Methods. The OPG concentration range (0, 40, 80, 100 ng/mL) was mentioned for the first time in the section “Results”.

Response: we have added.

Question 2:

Comment 2. Levels of LDH, IL-18 and IL-1β assay in the medium supernatant were not normalized to the cell content in each experimental condition. Plating the same number of cells will give different cell populations in the various experimental conditions – thus, some sort of normalization is needed (to total protein content, DNA content, …)

Response: before experiments, we quantified with BCA kit. We have added to the method.

Question 3:

Comment 3. SEM images (Figure 1) are really of poor quality – they do not allow the detailed description of the morphology and alterations described the authors. The same is observed with the light microscopy images – they do not allow to identify multinucleated cells. Authors need to prove that they have osteoclastic cells – i.e., performing immunostaining of the cytoskeleton and nucleus allows the visualization of the typical actin rings and the multiple nucleus. Even histochemical TRAP staining (a very basic staining) would provide useful information. And, why TRAP activity was not determined – it is a very basic biochemical method of an osteoclastic marker.

Response: SEM image has marked. we have added TRAP staining.

Question 4: In a previous work reported by the research team (ref. 7), authors used a similar experimental setup for the obtention of the osteoclasts (bone marrow from male Balb/cJ mice) and also analysed the effect of OPG (in a similar range - 0, 20, 40, 80 ng/mL, apparently, 24 h treatment) in the dead rate. In this past study, they used the term “apoptosis” (Q2 quadrant) but in the present work, with the same protocol, the term used was “pyroptosis” (Q2 quadrant). They analysed distinct underlying mechanisms for the cell dead rate – “Osteoprotegerin Induces Apoptosis of Osteoclasts and Osteoclast Precursor Cells via the Fas/Fas Ligand Pathway” (Ref 7) and “The mechanism of osteoprotegerin-induced mice osteoclast pyroptosis” (present study). Cells in Q2 quadrant may behave similarly in flow cytometry using Annexin V-FITC/PI kit (cells in late apoptosis /pyroptosis), and they can be distinguished by complementary information and underlying mechanisms. But relating the two studies using a similar protocol, OPG may be inducing both cell dead processes? Surprisingly, in the “Discussion” section, authors never mentioned this previous study, and discuss/relate the results of both works. Why these two studies were not discussed in an integrative way? 

Response: thanks for your question, this is a good question. Firstly, in pyroptosis study, many papers used FITC/PI to observe pyroptosis in Q2 quadrant [1]. According to apoptosis definition, the membrane integrity not destroy, Q3 quadrant cell was considered for apoptosis. However, in this study, we think apoptosis and pyroptosis were simultaneous, OPG treatment for different time, the occurrence rate of apoptosis and pyroptosis is not same.

[1] Chemotherapy drugs induce pyroptosis through caspase-3 cleavage of a gasdermin. Nature, 547, 99-103.

Question:

Comment 5. The Conclusions of the two studies reflect the obtained results:

-        Previous study (ref. 7): In conclusion, our investigations reveal that OPG promotes apoptosis of OCs and OPCs by activating the Fas/FasL pathway. In addition, OPG-induced Fas/FasL activation regulates osteoclast apoptosis using any cell-type-specific signaling pathways. Such pathways could provide novel designs of osteoclast-specific therapeutic interventions in bone disorders. Therefore, future research is required on the precise Fas-mediated intracellular signaling pathways in osteoclast apoptosis induced by OPG.

-       Present study: In this study, the osteoclasts were treated with OPG for 12 h; OPG could affect the survival of osteoclasts in a dose-dependent manner. In addition, OPG treatment induced pyroptosis of osteoclasts by up-regulating the transcription levels and protein expression levels of genes related to pyroptosis classical pathways such as ASC, NLRP3, caspase-1, and GSDMD. In all, this study preliminarily revealed the mechanism of OPG-induced pyroptosis of osteoclasts, providing a theoretical basis for the treatment of bone metabolic diseases.

 In a translational point of view - novel insights of osteoclast-specific therapeutic interventions in bone disorders – stressed by the two studies, what will be the difference in the clinical outcome?

 Response: thanks for your question. Firstly, the previous study (ref.7) cultured cells in the presence of 40 ng/ml RANKL (PeproTech, Inc), in this condition, BMMS can not absolutely induce mature osteoclast, it is mainly the precursor osteoclast, and OPG (0, 20, 40, 80) treatment for 24 h, therefore, it may be induced precursor osteoclast apoptosis. in this study, we obtained mature osteoclasts in the presence of 60 ng/ml RANKL (R&D, USA), (R&D RANKL activity is more sensitivity than Pepro Tech), and 100 ng/ml OPG treatment for 12 h. Therefore, in different conditions, OPG may play a different role in osteoclast. In addition, we think, after OPG treatment, apoptosis and pyroptosis may be simultaneous, low concentrations OPG induced osteoclast apoptosis, high concentrations OPG induced pyroptosis. Therefore, different concentrations OPG play a different roles in bone disorders interventions.

Round 2

Reviewer 1 Report

Dear authors,

I have just one minor suggestion on the manuscript:

Figure 1e: please use another color (or write the conc. below), since red color is difficult to read.

Author Response

Dear editor

      thanks for your question, Figure 1e: please use another color (or write the conc. below), since red color is difficult to read.

    We have solved this question. and the results part and conclusion have improved.

Reviewer 2 Report

Figure 2B, introduced by the authors in this revised version, does not add any information. Also the added text "We further stained with PI to observed the integrity of osteoclast membrane. As shown in Fig 2B, OPG significantly destroyed the integrity of osteoclast membrane", should be delected - because this is not true! Osteoclasts membrane is not visible in these images!!

The authors addressed only partially some of my comments:

Nothing was added in the manuscript concerning the two last comments. Specially, it seems very strange that nothing has been mentioned/compared/discussed in the present manuscript related to the results observed in the previous study concerning the same research topic (ref. 7).

Author Response

Question: Figure 2B, introduced by the authors in this revised version, does not add any information. Also the added text "We further stained with PI to observed the integrity of osteoclast membrane. As shown in Fig 2B, OPG significantly destroyed the integrity of osteoclast membrane", should be delected - because this is not true! Osteoclasts membrane is not visible in these images!!

Response: thanks for your question, we have deleted.

 Question: Nothing was added in the manuscript concerning the two last comments. Specially, it seems very strange that nothing has been mentioned/compared/discussed in the present manuscript related to the results observed in the previous study concerning the same research topic (ref. 7).

Response: we have added in discussion.